# Teacher Trainees' Perspectives on Remote Instruction for Multilingual Learners of English

Kandace M. Hoppin [1] , Gregory Knollman [1], Patricia Rice Doran [1,*] and Huili Hong [2]

1    Department of Special Education, Towson University, Towson, MD 21252, USA;
     khoppin@towson.edu (K.M.H.); gknollman@towson.edu (G.K.)
2    Peabody College of Education, Vanderbilt University, Nashville, TN 37203, USA
*    Correspondence: pricedoran@towson.edu; Tel.: +1-410-704-3891

**Abstract:** The COVID-19 pandemic prompted a shift to virtual learning across many countries and school systems. It is worthwhile to examine the specific ways in which this shift is significant to teacher trainees preparing to work with multilingual learners (MLs). Considering the perspectives of teacher trainees preparing to teach MLs offers an opportunity to identify the questions and concerns that they are likely to have upon graduation. Examining these perspectives can also help to identify ways that teacher trainees can use virtual and remote teaching approaches more constructively. This paper presents findings from a qualitative study of an educator preparation program focused on preparing trainees in content areas along with English to Speakers of Other Languages (ESOL), with a focus on the perspectives of teacher trainees who worked with MLs through virtual and remote modalities during the COVID-19 pandemic. The paper draws on data from an analysis of nine teacher trainees' response journals and course assignments, and includes themes identified from the teacher trainees' perceptions of virtual learning for MLs. The findings from the analysis revealed that teacher trainees emphasized the importance of establishing meaningful professional relationships in the virtual setting with their MLs, especially as a way to facilitate effective instruction and online classroom management. Participants also spoke about the importance of developing culturally responsive and sensitive instruction, and stressed the importance of engaging students and families in appropriate, linguistically accessible ways. Implications for future virtual instruction as well as teacher preparation are also discussed.

**Keywords:** multilingual learners; English learners; English language instruction; teacher trainees; virtual learning; remote teaching

## 1. Introduction

Beginning in March 2020, several countries turned to remote learning in response to the COVID-19 pandemic. An estimated 107 countries implementing national school closures related to COVID-19 found themselves rapidly scaling up technology to address the needs of students who were homebound because of the pandemic (UNESCO 2020; Viner et al. 2020). During the first year of the pandemic, many of the largest urban districts within the United States provided at least partial online schooling (Stuart et al. 2021). Even if schools have resumed in-person learning, teachers and school leaders must consider how to use the technology to address learning loss from this shift to remote learning (Korkmaz and Toraman 2020).

Teachers must be prepared to work with diverse populations, particularly students learning English, referred to in this paper as multilingual learners (MLs). In doing so, teacher trainees must be ready to provide continuous, high-quality English instruction for students who have already experienced substantial stress and educational disruption over the past year. Teacher trainees graduate and take positions in districts where challenges related to access to technology and internet connectivity continue to exist (Lockee 2021).

Together, these circumstances create new pressures on teachers of English, but also present unique opportunities and ways to leverage teachers' skills to provide robust culturally and linguistically responsive learning opportunities in virtual as well as face-to-face modalities. Teacher trainees must remain aware of the complex ways in which students may interact with information and the variety of learning experiences that can support them in building knowledge (Kalpana 2014). Teachers may play an important role in structuring and directing these interactions and the student learning that results from them. Student actions and interactions are influenced not only by students' growth and development but also by the social environment and culture of the classroom (Kalpana 2014). Competent teachers can structure classroom routines, activities, and experiences to provide student-centered learning activities to help their students engage in meaningful learning experiences. Within the online learning environment, as well as the face-to-face one, teachers may influence the dynamic, social, and interactive nature of language instruction and learning, which is supported through interaction, communication, and collaboration (Carwile 2007). Accordingly, it is helpful for teacher trainees to possess a strong understanding of these complex dynamics around learning environment, learner background, language and social interaction, and student well-being.

Considering the perspectives of teacher trainees who intend to serve as English language teachers provided an opportunity to identify the questions and concerns that they are likely to have, as well as highlight potential ways that they can use virtual and remote teaching approaches in transformative ways to support their MLs. This paper presents findings from a qualitative examination of the perspectives of teacher trainees completing an English to Speakers of Other Languages (ESOL) preparation program in the Mid-Atlantic region of the United States, regarding working with MLs via virtual and remote modalities. The paper presents findings from ESOL teacher trainee response journals and course assignments focusing on their beliefs, experiences, and perceptions about MLs and virtual and technology-assisted learning. Last, the paper identifies implications for current and future practice, including opportunities for strengthening teacher preparation and English language instruction. The research questions for this study were:

- How do teacher trainees describe their experience as a learner within a virtual experience in the COVID-19 pandemic?
- How do teacher trainees of MLs perceive the challenges of virtual learning, especially considering students' experiences in the COVID-19 pandemic?
- How do teacher trainees of MLs perceive the benefits of virtual learning, especially considering students' experiences in the COVID-19 pandemic?
- What strategies or approaches do teacher trainees of MLs appear to find promising or useful for supporting their MLs in the process of virtual learning?

*1.1. COVID-19 Pandemic and Virtual Teaching of English*

As school systems around the world shifted to remote learning during the COVID-19 pandemic, the rapid changes in educational delivery required adjustment and adaptation for teachers of all students at all levels but posed specific challenges for language teachers (Moser et al. 2021). For example, many nonverbal and paraverbal means of learning language, such as tone, social context, gesture, and body language, can be challenging to convey over video chat or asynchronous teaching modalities. Established techniques such as Total Physical Response (TPR) are difficult or impossible to implement remotely. Moreover, students who rely on frequent, verbal interaction to practice oral language skills may be hampered by the limits of face-to-face time that are often present even in synchronous virtual instruction. While there were many challenges to language acquisition through online learning, virtual and remote modalities offered new and potentially transformative ways to engage MLs and their families (Raghavendra and Chikkala 2020). Technology allowed families to connect with teachers and even with other families; chat and video conferencing allowed students to practice skills in real time with peers even outside the

classroom; and virtual modalities allowed teachers to be present in new and unique ways for students' oral language practice, group work, or dialogue with individual students.

## 1.2. Deployment of Online Learning and Teacher Readiness

Prior to the pandemic, higher education made slow and steady progress in a transition to online teaching and learning by adopting varied pedagogical approaches to teaching and a range of technologies resulting in widely varied faculty attitudes of readiness to adopt these tools (Howard et al. 2021; Martin 2019). In one such study of teacher perceptions prior to the pandemic, Gurley (2018) surveyed faculty perceptions of the adoption of these online modalities and found those instructors completing certification courses to teach in blended or online learning environments held a higher perceived outlook on their success in teaching online compared to colleagues who received more general training. When considering faculty engagement in the adoption of new technologies within higher education, Bennett (2014) described the transition as an emotional process that required strategies to manage the challenges of learning new tools while also providing quality learning experiences to the students.

The pandemic resulted in a dramatic shift from teaching in person and the gradual adoption of new technologies to the rapid deployment of a fully online curriculum. College students expressed challenges in accessing reliable internet connectivity, finding a quiet space to complete online learning, concerns with finances, and fears of losing social connections with peers, faculty, and the college community (Gonzalez-Ramirez et al. 2021). Teachers in many content areas were required to redesign curriculum and instruction for remote delivery. These changes in instructional design and delivery reshaped the education landscape almost instantly with profound changes to teaching and assessment (Middleton 2020). The shift in content delivery impacted teacher perceptions differently across online learning environments (Marshall et al. 2020) and further illuminated inequities in district responses to online instruction (Hall et al. 2020). During the pandemic, Scherer et al. (2021) developed a profile of teacher readiness, which was designed to measure the teachers' preparedness to rapidly deploy technologies required to support online instruction. This profile was designed from the results of a survey that examined both instructors' personal readiness to use online instruction and the readiness of their campuses to support online teaching and learning. Among survey respondents, those who held poor self-beliefs in their ability to adapt to online teaching and learning also held more negative perceptions of their institutions' readiness to support their transition to online learning. Furthermore, the faculty members who had prior experience with online teaching and learning held higher perceptions of their readiness to adapt to new technology use.

Aside from teacher perceptions of the adaptation to online teaching, setting up instruction through virtual learning brought its own unique challenges and possibilities, which are particularly relevant to current and future teachers of English. For example, the focus on technology and technological proficiency is often less explicit within English language pedagogical models, and often treated as a separate set of competencies rather than a foundational skill. Furthermore, as Altavilla (2020) highlighted, technology represents an under-addressed area both in the curriculum and services provided to MLs. Accordingly, even when teachers utilize technology, they may not possess the pedagogical knowledge or practical experience to leverage it to maximum effectiveness.

## 1.3. Student Well-Being and Success While Engaged in Virtual Learning

Teachers and researchers have reported increased concerns about the well-being and success of students during virtual learning, particularly MLs (Catalano et al. 2021; Cushing-Leubner et al. 2021). Without intention, focused exploration of barriers and opportunities specific to virtual learning, educators run the risk of reproducing existing inequalities in a face-to-face classroom when they shift to a virtual space (Green and Tolman 2019). In a review of recent literature, Bartley (2021) identified factors including the following to be important to creating positive virtual learning experiences: "connections" and "relation-

ships" (p. 1) with students as well as family members, "assets-based" approaches (p. 1), and a strong awareness of students' social and emotional status and needs.

Sayer and Braun (2020) highlighted the way in which the shift to remote learning had particularly challenging ramifications for MLs, including technology access, abrupt discontinuation of access to sheltered content, and reduced opportunities for practice of oral language skills that are often crucial to developing overall language proficiency. However, these very real difficulties were also accompanied by the "silver lining" of increased opportunity for connection (p. 4) between educators and families using technology, an opportunity that may persist in face-to-face learning where technology is utilized appropriately.

The ability to leverage technology has potential positive impacts for instruction beyond the COVID-19 pandemic, as technology can provide a powerful way to address instructional barriers. Pre-pandemic, Smith and Stahl (2016) identified a need for increased accessibility and emphasis on Universal Design Learning (UDL), as an increased number of students access online and virtual learning opportunities. UDL is a framework with origins that predated the sudden shift to fully online experiences during the pandemic by several decades. The underlying framework principles emphasized the importance of promoting instruction that includes multiple means of representation, expression, and learner engagement (Edyburn 2005; Meyer et al. 2014; Rose 2001). The term was first defined in two pieces of federal legislation within the 2004 reauthorization of the Individuals with Disabilities Education Act (IDEA) and the Assistive Technology Act of 1998 and applied to the design and delivery of products and services, including assistive technologies that improve access to instruction and use of content by a wide range of people (Edyburn 2005).

Through research conducted during the pandemic, Flanagan and Morgan (2021) found that integration of UDL into instructional practices can help all learners, particularly those with disabilities, to be successful. Similarly, Basham et al. (2020) emphasized the importance of focusing on universal design to ensure all students can access learning opportunities and be successful. Related to the challenge of effectively supporting all learners, Chang (2021) highlighted the challenges of maintaining student privacy in the virtual setting, including both legal and compliance issues and issues of professionalism, interaction, and confidentiality in a virtual setting.

*1.4. Student–Teacher Relationships in Virtual Learning*

Even before the COVID-19 pandemic, researchers had begun to address the importance of intentional relationship-building in online learning environments for school-aged learners. Borup et al. (2013) found that with deliberate adjustment in practices, teachers could build strong relationships with learners in a virtual setting characterized by caring. Likewise, Drysdale et al. (2014) found that structured programs focused on mentoring and supporting students, including an emphasis on relationships as well as instruction, could facilitate students' success and teachers' well-being. Martin (2019) described strategies teachers can use to successfully engage with students and build relationships in virtual settings. In reviewing the impact of virtual and remote learning on MLs during the pandemic, Bartley (2021) reported relationship-building to be important for students' success. Likewise, in examining teachers' experiences during the COVID-19 pandemic, Miller (2021) found that students engaged in remote learning during the pandemic brought unique concerns to the classroom. However, teachers could provide support by acting as "warm demanders" (Miller 2021, p. 115) to encourage continued growth and learning, as well as ensuring accessibility and offering socioemotional support. Hamilton et al. (2021) also explored teacher practices and family engagement during the pandemic to identify areas for focus and emphasis in school practices. Teachers worked quickly to shift to remote learning, but reported a need for more detailed strategies to connect with students and maintain engagement, as well as using technology effectively.

## 2. Materials and Methods

### 2.1. Methods

This study was conducted by a research team consisting of several collaborators and authors. The team included the instructor of the course in which participants completed journal assignments, the lead interviewer for qualitative interviews, and two team members who investigated the relevant literature and were involved in all levels of data analysis. The research team used a descriptive–interpretive qualitative methodology (Elliott and Timulak 2021), which relied on qualitative analysis of teacher trainees' reflective journals, as well as interview data, exploring the trainees' lived experiences with virtual learning and instruction of MLs (Brinkmann and Kvale 2014). The research design of this study was based on an interpretivist paradigm, and the research team used a basic interpretive design (Creswell and Creswell 2018; Merriam and Tisdell 2016) to explore the experiences of teacher trainees preparing for and engaging in virtual instruction. The overall purpose of descriptive–interpretive qualitative design was to understand how individuals make sense of their lives and experiences (Merriam and Tisdell 2016; Patton 2015). Highlighting the experiences of teacher trainees preparing for and engaging in virtual instruction of MLs through a qualitative lens helped to expand the potential for understanding the complex issues related to language, virtual instruction, and successfully supporting future educators.

### 2.2. Participants

Data were collected from a total of nine participants for this study (Table 1). Participants were members of a cohort of undergraduate teacher trainees enrolled in a large United States university's College of Education completing elective coursework in ESOL in addition to their primary area of certification in order to gain endorsement in ESOL upon graduation. The cohort of teacher trainees from which this group of participants was drawn included trainees from varied fields, including Early Childhood, Elementary Education, Secondary Education, Special Education, and Elementary Education–Special Education (dual certification). All participants had completed at least one course in multicultural and multilingual education as part of this program, and were enrolled, at the time of data collection, in two additional courses with an ESOL focus. One of these courses focused on methods for teaching MLs, and the other, in which data for this study were collected, focused on assessment for MLs. Throughout this course, the teacher trainees engaged in virtual weekly tutoring and instruction of a small group (2–4 students) of MLs.

Purposeful sampling (Creswell and Creswell 2018; Patton 2015) was used to select participants whose experience would be particularly relevant to the topic of the study. Participants were selected based on the following criteria:

- Completion of journal prompts and interview questions that focused on virtual learning and perceptions of MLs: All participants participated in the journal prompts as part of an elective course assignment, including informed consent. All but one participant consented to complete at least one 30–60-minute semi-structured interview in which they described their experiences and perceptions with respect to cultural and language diversity, virtual instruction, and instruction or intervention for MLs

- Phase of the professional program: All participants were enrolled in a selective teacher education major at a well-established college of education located within a large university. Participants had all completed a pre-professional year of coursework and were in the process of preparing for a professional year involving part-time and full-time school-based internships. All participants had completed virtual fieldwork and participated in virtual instruction in the pre-professional year as a result of the COVID-19 pandemic and the associated shift to remote learning

- Major field of study: The study focused on participants majoring in early childhood, elementary, secondary, or special education, who were members of a cohort obtaining concurrent eligibility for endorsement in K-12 ESOL, the state's certification track for English language teachers of multilingual learners. All participants had completed three credits in ESOL and were in the process of completing six additional credits in

ESOL toward this endorsement at the time of data collection, providing them with a common knowledge base and set of reference points regarding the instructional and language needs of MLs

- Prior experience with a foreign language or as MLs: Participants had the opportunity to identify whether they considered themselves multilinguals, although responses in this regard were not used to exclude or include participants. Participants' status with respect to this category is indicated in Table 1.

**Table 1.** Characteristics of participants.

| Participant # | Major | Self-Identified as Language Learner |
|---|---|---|
| Participant 1 | Elementary Education and Special Education (ESOL endorsement) | No |
| Participant 2 | Elementary Education and Special Education (ESOL endorsement) | No |
| Participant 3 | Secondary Special Education (ESOL endorsement) | No |
| Participant 4 | Elementary Education and Special Education (ESOL endorsement) | No |
| Participant 5 | Elementary Education and Special Education (ESOL endorsement) | No |
| Participant 6 | Elementary and Middle Grade Special Education (ESOL endorsement) | No |
| Participant 7 | Elementary Education and Special Education (ESOL endorsement) | Yes |
| Participant 8 | Early Childhood Education (ESOL endorsement) | No |
| Participant 9 | Early Childhood Education (ESOL endorsement) | No |

### 2.3. Data Collection

Data were collected from reflective journal entries that each participant submitted for a summer course on assessment and instruction of MLs, taught by one of the research team members. Participants were expected to complete eight, one-page journal reflections throughout their course, three of which were utilized for this study, as these selected reflections related most specifically to virtual learning and instruction of MLs (the other five prompts were not relevant to virtual learning or teaching). Each of the nine participants in this study responded to all three of the selected reflective journal prompts, for a total of 27 collected journal responses. The average length of the journal responses from participants was 250 words. Reflective journals included in this study were in response to the following prompts:

- Prompt 1: Describe your experience thus far with virtual tutoring. How would you assess and describe your student/s' language proficiency? How would you describe their learning strengths and needs? If you have not yet begun virtual tutoring, please explain how you plan to assess these items when you do begin?
- Prompt 2: Take a moment to reflect on the experience of virtual learning and teaching. What do you think is challenging or different about virtual learning, especially for multilingual learners (MLs)? Are there any benefits or upsides to MLs regarding virtual learning?
- Prompt 3: How confident do you feel about your ability to provide instruction within a virtual platform? Do you have greater confidence in your ability to teach students with certain needs within this environment? What practices will you utilize in your own teaching practice to support students and families in accessing technology and virtual experiences? If so, explain.

In addition to the reflective journal response data, one-on-one semi-structured interviews with one of the research team members were conducted with each participant upon completion of the course. The research team devised a semi-structured interview protocol

to serve as a guide to help ensure consistency among each participant and to allow individual perspectives and experiences to emerge (Patton 2015). Interviews were conducted by a member of the research team, who was familiar with the course expectations and journal reflections, but who did not teach the course from which the data were collected. Interviews focused on participants' background knowledge, experiences, and perceptions regarding virtual learning and instruction of MLs. Participants responded to prompts and follow-up probing questions regarding their experiences with MLs, experiences with prior coursework or fieldwork related to supporting MLs, and perceptions of the virtual learning process and experience. Participants completed informed consent at the start of their professional development experience and, for participants completing interviews, reviewed consent procedures orally again during the interview with the research team member conducting their interviews. All procedures were approved by the researchers' university Institutional Review Board (IRB), and informed consent was obtained from all participants in the study. Interviews lasted between 30 and 60 minutes and were audio-recorded and then transcribed.

The addition of interview data allowed the team to capture not only participants' direct responses to the journal prompts but also their perspectives on the lived experience of being a teacher trainee participating in, as well as preparing for, virtual learning and instruction (Brinkmann and Kvale 2014).

*2.4. Data Analysis*

Journal response data and all interview transcripts were stored in a password-protected online database to which each member of the research team had access. Participants' data were also imported and organized using NVivo, which is a qualitative data analysis application. The research team triangulated the data (Patton 2015) from the reflective journal responses and the transcribed interviews to gain a nuanced and multi-faceted perspective on the trainees' experiences and perceptions.

The researcher team coded the reflective journal responses and the transcribed interviews as a group that met virtually throughout the coding process. In order to create a valid and reliable qualitative study, the researchers used memoing and created an audit trail of the research steps taken by the team to preserve the integrity of the participants in the study (Creswell and Creswell 2018; Merriam and Tisdell 2016; Miles et al. 2020). The research team reviewed all transcripts and journal entries together, and engaged in multi-level coding (Saldaña 2015) in which key codes were first identified, then consolidated into categories, reviewed, and interpreted as general themes were identified within and across topics and questions.

The first step of analysis involved three members of the research team coding a set of three participants' data, including those three participants' reflective journal responses and transcribed interviews, independently. After this initial round of coding, the research team came together as a group to determine how they understood and interpreted similar themes and constructs from the participants' data that were analyzed, and then generated an initial codebook. During the discussion, the team established consensus in defining each code from the journal and interview data. The research team worked to complete a second round of coding as a group of the full data set. The team discussed and arrived at a consensus on the overall themes that emerged. Simultaneously, the team individually wrote memos and notes that allowed for elaboration on coding processes and reasons for decisions. Memo writing helped clarify emergent categories and themes during the coding process, and gained consensus as a group on the findings.

*2.5. Themes and Subthemes Aligned with Research Questions*

Four overarching themes occurred consistently across participants' responses in journals and interviews: relationships, engagement, flexibility, and appropriate use of technology. Further analysis by the research team revealed that teacher trainees' insights

about MLs and virtual learning could be grouped into several subthemes aligned with the research questions and included in Table 2.

**Table 2.** Subthemes aligned with research questions.

| Research Question #1: Experience as a Learner within a Virtual Environment | Research Question #2: Challenges of Virtual Learning | Research Question #3: Benefits of Virtual Learning | Research Question #4: Strategies and Approaches |
|---|---|---|---|
| ○ Flexibility<br>○ Adjustment<br>○ Relationship-building<br>○ Benefits to virtual teaching and technology | ○ Technology challenges<br>○ Fidelity and rigor of assessment<br>○ Language barriers<br>○ Relationships<br>○ Engagement and management<br>○ Differentiation for language learners<br>○ Professional practices<br>○ Time and effort | ○ Student comfort<br>○ Universally designed instruction, accessibility, individualization<br>○ Integration of technology<br>○ Privacy and personal connections with learners<br>○ Virtual relationship-building<br>○ Streamlined planning processes | ○ Parent and family communication<br>○ Confidence<br>○ Use of tech tools and applications<br>○ Adapting and reflecting on practice<br>○ Relationship-building<br>○ Positivity |

## 3. Results

Findings are summarized below with respect to each research question; a general discussion of subthemes and results follows.

### 3.1. Question 1: How Do Teacher Trainees Describe Their Experience as a Learner within a Virtual Experience in the COVID-19 Pandemic?

The teacher trainees in the study revealed varied perspectives on their own experience of virtual learning. All had completed virtual learning as university students during the COVID-19 pandemic. All had completed at least some virtual teaching via a one-on-one tutoring program offered by their college in conjunction with certain courses. Participants' responses indicated the following as key elements that they perceived in virtual learning:

#### 3.1.1. Adjustment and Flexibility

Among the perspectives shared on the transition to virtual learning, participants' responses focused on the sudden adjustment or adaptation needed to acclimate to the process of virtual instruction. In an interview, a participant described virtual teaching as "really, really weird" at first, and expressed concern about approaching parents through online modalities or connecting with them in culturally sensitive ways without the benefits of in-person interaction: "I'm still learning that and … I hope … in the future, we can still learn." Another participant described, in a journal response, feeling uncertain about how to deliver online instruction as a result of being a student still figuring out how to learn virtually: "I felt extremely uncertain about my ability to teach and provide instruction to students. I felt like I was still figuring out how to learn virtually myself so teaching on that platform to others was a bit intimidating to me."

Virtual learning also offered flexibility, particularly during a challenging time for many families and individuals adjusting to COVID-19 restrictions and circumstances. One participant commented in an interview: "I would have gotten more out of it, but it was also very beneficial … because then I could be home for my coursework." This flexibility, though, necessitated some adjustment, as described above, as the process of learning virtually was not intuitive.

### 3.1.2. Relationship-Building

Seven participants emphasized the value of continued emphasis on personal relationships through referencing topics such as student engagement, connections with home, connection with students' cultures, and family communication. A participant indicated valuing interaction and individual meetings extended to them by professors and indicated this was important to carry such practices forward into their own teaching. One participant stated in an interview: "I feel like a lot of professors were … hard on themselves … the important thing is that [students] learned … the important thing is … they feel like they matter and they matter to you." Ensuring students feel this connection, though, can come at some cost to teachers, as this participant also shared: "You really have to go out of your way more, in a virtual environment, to make sure you're making those … connections."

### 3.1.3. Need for Differentiation, Particularly for MLs

All participants identified at least one challenge related to virtual teaching and learning, including their own experience as learners as well as their students' experiences as multilingual learners. One participant shared in an interview: "Keeping the engagement for both students was difficult at times." One participant reflected in a journal entry: "It is hard to help the students directly, they will have to explain their problems or show their paper to the camera … Being told to explain a problem that you do not understand is very hard, especially if that language is not your native. I feel bad when we ask students to tell us where they are stuck."

### 3.1.4. Benefits of Virtual Learning

All participants expressed uncertainty, ambivalence, or even concern about virtual learning, but seven participants also expressed optimism or positive ideas about aspects of technology, including the ability to engage students, bridge gaps or involve families. For example, one participant reflected in an interview: "I'm not sure anybody knows the long-term … developmental implications that this will have on students, but I feel like it's only gotten better and better and better because [of] the amount of technology that we have." It appeared that the ability to practice and improve over time also could be helpful; in the words of one participant, writing a journal entry: "We have been doing this type of tutoring since last semester and I feel like it is getting easier. I am slowly gaining confidence when it comes to speaking and teaching the students."

### 3.2. Question 2: How Do Teacher Trainees of Multilingual Learners (MLs) Perceive the Challenges of Virtual Learning, Especially Considering Students' Experiences in the COVID-19 Pandemic?

All participants identified at least one challenge to virtual learning, both as they related to their own experiences as students and, primarily, as those challenges related to their roles as a virtual teacher in teacher training experiences. Participants in this study had not experienced full-scale virtual classroom instruction with students but had instead participated in fieldwork focused on individual or small-group tutoring. Participants' concerns about the challenges of virtual learning can be grouped into several main categories described below:

### 3.2.1. Technology Challenges

Technology challenges such as internet connection, difficulty in communicating students' ability to find a learning environment, and difficulty of access were referenced by four participants. One participant described their experience in a journal entry: "The virtual learning process has been very difficult. As a student there were some moments where I had internet trouble, and this became worse once we started the tutoring process … when I have trouble with the computer my stress level just skyrockets." Challenges with technology also extended to concerns about the ability of different age groups to utilize technology successfully; another participant wrote in a journal entry that it was "a little hard to provide in-depth instruction, especially for younger children, because of [the need to minimize] their screentime."

### 3.2.2. Quality of Instruction

All participants reported some concerns around the quality of instruction, whether that involved addressing individual questions, monitoring student learning, ensuring students were focused, or addressing language barriers without in-person social cues. Informal assessment emerged as one area of concern or potential difficulty. One participant, for example, commented in a journal entry that "it can be difficult to see if a student is confused or not when you are trying to get through a lesson."

Concerns about the quality of instruction also extended to issues specific to language learners. One participant commented in a journal entry, "It is hard to help the student directly … being told to explain a problem that you do not understand is very hard" for MLs in particular. This participant also referenced the difficulties in providing feedback on student work via remote learning, as students would often be required to hold their work up to the computer camera in the absence of scanning and uploading it. As one participant put it in a journal entry: "It can be a little harder for [MLs]/Virtual learning makes it a little difficult to get individualized help, in a class full of students." Logistical challenges made providing help more complex in a face-to-face traditional setting, as the same participant explained: "[Y]ou have to step into a breakout room", requiring a student to be removed from peers for the duration of the help session.

The interactive nature of language learning and teaching posed some difficulty to one participant, who commented in a journal entry: "I think [MLs] thrive off of interaction and physical representations and examples. In a virtual space, those things can be hard to accomplish." These concerns indicated general ambivalence or hesitancy around the best way to provide high-quality, appropriate instruction for MLs in a virtual setting; this same participant described their own view of virtual learning as being "a little hesitant" as a result of these challenges.

### 3.2.3. Time and Effort

Time and effort in planning and locating materials was a focus for at least two participants. One participant referenced the time required to locate materials: "[It] definitely wasn't easy but it also wasn't terrible in the sense that there was a lot more resources, just finding them, sometimes it was [not easy]." One participant referenced the increased time and adjustment needed to teach in a virtual setting, describing the experience in an interview, "It was definitely a learning curve for me."

### 3.3. Question 3: How Do Teacher Trainees of MLs Perceive the Benefits of Virtual Learning, Especially Considering Students' Experiences in the COVID-19 Pandemic?

In addition to detailing some challenges and drawbacks of online learning during the pandemic, seven participants also shared insights about benefits of virtual learning. These benefits were apparent in several areas, including the ability to respond to students' unique needs, differentiate for students, build relationships with students and families, and plan and deliver instruction efficiently using technology. Below is a summary of participants' observations in each of these areas:

### 3.3.1. Individualization and Differentiation for Unique Student Needs

Despite ambivalence about students' needs being met in virtual settings, three participants noted ways that virtual or remote teaching could support differentiation and individualization: "When we are in class, Zoom gives us the opportunity to talk individually and there is no talking over [students]", as one participant wrote in a journal entry. Five participants referenced increased availability of materials for teachers, allowing them more options to share differentiated or additional material with students: "I think the online environment can be beneficial to teach students with certain needs because there are more resources readily available to them like translated versions of books and spellcheck when typing", as one participant reported in a journal entry. In an interview, another participant stressed that finding appropriate and engaging materials "was a little bit easier

online" than in face-to-face instruction. One participant noted in a journal entry that virtual learning could be aligned to UDL in that it allowed students to "see and access pictures that can be helpful to [MLs]." Virtual learning was also seen as helpful by this participant because it "leaves room for more technology such as software that can be used as tool, and videos, where some videos come with subtitles." One participant also noted that videos could be leveraged to show information in multiple ways or to pause and replay after addressing student questions, although another participant wrote in a journal entry that showing videos was helpful but sometimes diminished their opportunities to interact or engage with students: "I found myself showing a lot of examples and having the students watch instead of engaging with me. I think this can be even more challenging with Els in the class."

### 3.3.2. Virtual Strategies for Relationship-Building

Technology seemed to foster more effective or meaningful connections with families: "I found I was able to connect more with my students", in the words of one participant as recorded in a journal entry; for this participant, "being more personable with the students and families" was seen as a distinct experience over other modalities of instruction. It appeared that virtual instruction also allowed some opportunities for increased interaction with families. For example, one participant wrote in a journal entry: "One of the advantages to online learning is the amount of interaction the teacher has with the students." This personal interaction led to deeper relationships with students as well as more frequent or meaningful contacts with parents for this participant: "In my personal experience I found that I was able to learn more about my students on a personal level and I was also able to interact with the parents more too. I thought those interactions were beneficial because it allowed me to tailor lessons to what my students are interested in, and I was also able to communicate with the parents and make sure the learning process was getting carried over to the home environment."

### 3.3.3. Time and Planning Innovation

While participants spoke of the increased time and effort required to plan and implement virtual instruction, they also referenced efficiencies that accompanied their use of virtual teaching strategies. As one participant stated in a journal entry, "There are so many resources I can use [to complete a lesson] and I can just add [them] to the PowerPoint." The same participant also wrote that "if I don't have the answer to a question I can just look it up quickly without forgetting about it." This finding was mirrored in the references that participants made to their increased ability to provide captions, recording, and integrate technology seamlessly into instruction to address student learning needs

### 3.4. Question 4: What Strategies or Approaches Do Teacher Trainees of MLs Appear to Find Promising or Useful for Supporting Their MLs in the Process of Virtual Learning?

### 3.4.1. Parent/Family Communication

Five participants referenced the benefits of family involvement in the tutoring session or the benefits of sharing information with families in some way. These varied benefits included establishing meaningful partnerships, sharing information with parents, and receiving crucial information from parents, often easier to do when technology facilitated communication with parents. One of the participants shared in an interview: "You get … the additional time with the family … and you also get the family more involved if they're able to [be]." This participant also went on to acknowledge that some families might not have the ability to participate directly in virtual or remote instruction: "It actually puts a large amount of stress on certain families, this whole virtual thing." For another participant, family communication allowed teachers to understand family stresses and obligations: "We had a parent apologize that her son had missed sessions because they were having some family issues." In a journal entry, a participant acknowledged the benefits of securing buy-in and sharing information with family members: "Instead of giving students the

resources and leaving it alone, we should make sure the student and parent understand the resources and the benefit of them."

### 3.4.2. Use of Appropriate Tools and Techniques, including Technology

Five participants referenced the use of websites, games, online scavenger hunt activities, apps, and quiz sites such as Kahoot! and tools embedded within Zoom (breakout rooms or screensharing of PowerPoint presentations, for example) as ways they could meet students' needs using technology in the virtual setting. These tools were seen as particularly important as ways to build or maintain student engagement and interest for MLs. Two participants referenced the importance of interactive games or activities in their journal reflections. One participant identified some weaknesses in understanding and using appropriate techniques but set a goal in their interview: "I want to know a couple of more strategies . . . " Another participant, in a journal response, noted growth in their skills and confidence in this area: "I have grown in my confidence to teach students with certain needs in the virtual environment . . . I still need to adapt my practices based on their skill levels."

### 3.4.3. Relationship-Building

All participants emphasized their positive perceptions of relationship-building practices, defined broadly as encompassing communication, culturally responsive practices, personal interactions with students, or emphasis on engagement. These practices extended to students as well as families. For example, one participant stated in an interview: "Sometimes we allow them to teach [students] like different words from their culture and be able to connect with them." Another participant referenced the value of concluding lessons on a positive note, and two participants described using different techniques or strategies, such as brain breaks, to maintain student engagement. In this respect, these participants evidenced awareness of some of the same strategies and approaches prioritized in traditional face-to-face instruction, particularly for inexperienced teachers.

## 4. Discussion and Limitations

### 4.1. General Themes across Research Questions and Interview Prompts

Several themes surfaced across questions and categories, thus indicating participants' interest in or focus on these concepts across different areas of consideration.

- Across topics and questions, participants returned to the theme of relationship-building. This theme encompasses relationships with multilingual students and with families. Participants expressed both challenges to relationship-building posed by the virtual environment and opportunities offered by virtual interaction that were not consistently present in face-to-face or traditional learning interactions. Engagement with families was expressed to be complex; at-home, virtual learning allowed participants to see and interact with families in some more authentic ways than school-based interactions allowed, but the at-home, virtual setting also created new challenges and barriers for engaging families, across cultures and languages, who might be pre-occupied with their own work or other pandemic-related priorities. This focus by participants is consistent with findings from the literature, including pre-pandemic research by Martin (2019) emphasizing the importance of relationships, as well as findings by Miller (2021) and Bartley (2021) regarding the particular value of relationships during remote learning, and particularly for MLs who might be considered to be at risk.

- In addition, participants expressed concern about their ability to deliver effective instruction, conceptualized broadly as related to engagement, student learning outcomes, and performance to their ML students. Within the area of effective instruction, participants emphasized considerations such as maintaining the engagement of students, gathering accurate and reliable informal assessment data, ensuring students were learning, and finding effective strategies for use in the virtual setting.

- Closely related to effective instruction, participants voiced the importance of flexibility. This included responsiveness to new needs articulated by students and families, willingness to change course in the midst of a lesson, seeking out and adopting new technologies or strategies appropriate for the virtual setting, or responding in real time to student questions. Participants both identified challenges in providing or modeling flexibility and articulated ways that a virtual setting could enhance teacher flexibility.
- These themes are both related to literature regarding the importance of UDL as a framework for designing and delivering instruction that supports MLs' unique needs. Again, this topic was widely discussed prior to COVID with extensive research documenting the value of a UDL-based approach to support language learning, as well as providing an open and accessible classroom environment. Research conducted during COVID corroborated this position (Basham et al. 2020; Flanagan and Morgan 2021). In this study, consistent with this use of the UDL framework, participants accorded particular value to flexibility, interaction, and ensuring accessibility for MLs, particularly when aware of the challenges that a virtual environment could pose to student learning.
- Finally, participants described the importance of appropriate, innovative, and useful ways of integrating technology into instruction. This included professionalism in the use of technology, as evidenced by the discussion of student privacy on the part of one participant; it also related closely to the concerns articulated by multiple participants around the selection and use of engaging, innovative apps, websites, or other digital resources for students. Last, participants emphasized the importance of technological proficiency; barriers such as using the wrong materials or having a poor Wi-Fi connection could pose significant problems in delivering real-time instruction to students. These challenges are not limited to classroom environments, as individuals of all backgrounds and ages may experience a lack of access to technology, networking, or materials. However, they have particular relevance to MLs, whose learning is often reliant on prompt, real-time feedback and high-quality communication.

*4.2. Pedagogical Implications*

Many of the insights gleaned from participants were focused on the practicalities of learning and teaching in virtual, technology-assisted modalities. As such, they have relevance for pedagogy moving forward, whether schools continue to provide virtual instruction or transition back to entirely face-to-face models.

Among these implications is the continued relevance of the UDL framework when educators prepare to deliver instruction using virtual or even in-person, technology-assisted platforms. A UDL-based approach (Basham et al. 2020) can provide teachers with guidance and a nuanced understanding of how to provide flexibility, encourage all learners to tackle challenging tasks, and individualize tasks where necessary. In addition, UDL provides a powerful perspective on minimizing barriers and increasing access, tasks that may in some ways be facilitated by technology, such as real-time captions, recording, and playback, or access to personalized digital resources. Whether educators are teaching in exclusively virtual modes or providing in-person instruction, such flexibility and routes of access can be leveraged to provide maximal opportunities for learning, particularly for students with unique language-learning or disability-related needs.

In addition, while virtual modalities may make it more challenging for teachers to build relationships, such relationships continue to be critical. Strategies such as one-on-one conferences, personalized connections, and games, as mentioned by participants, can all offer ways to build strong connections with students across virtual or face-to-face platforms. In addition, technology such as video conferencing and real-time messaging can make it easier to support student learning by forging connections with families.

Participants referenced, in multiple ways and at multiple points, the potential of virtual learning for individualization and pedagogical flexibility. Such potential can be harnessed in face-to-face settings as well, where teachers can supplement effective and engaging

face-to-face instruction with personalized technology, individually selected activities, and opportunities for structured review with tools such as video recordings and web-based resources. Approaches that utilize the benefits of technology and virtual learning can be useful supplements for in-person learning, especially when thinking about providing out-of-school resources or tailored interventions. It may be useful for educators to consider ways to continue utilizing these features in their in-person classrooms as well.

The findings of this study illustrated the complex ways in which teacher trainees of MLs perceive the process of virtual instruction, its challenges and benefits, and their role as emerging professionals utilizing practices geared toward positive outcomes for students. Multiple participants emphasized the importance of personal connection and relationships in the virtual setting with their MLs, especially as a way to facilitate effective instruction and online classroom management. This aligns with findings from current research, including pre-COVID work by Borup et al. (2013) and Martin (2019) and preliminary post-COVID studies by Miller (2021), among others. These findings emphasized the importance of relationships in all settings, whether traditional or virtual, but also highlight the particular importance teacher trainees may place on relationships with their culturally or linguistically diverse families in a remote or virtual setting. Participants also spoke about the importance of culturally responsive and sensitive instruction, stressing the importance of engaging students and families in appropriate, linguistically accessible ways and maintaining students' engagement in the classroom setting. In this respect, participants evidenced awareness of some of the same concerns that surface in traditional face-to-face instruction, particularly for inexperienced teachers seeking to effectively support their multilingual learners.

### 4.3. Limitations

This study has several limitations. First, this study explores findings from a small group of participants and a relatively small set of data sources. While relatively small cohorts are not uncommon in qualitative research, it is important to keep in mind that they are not necessarily representative due to size. Similarly, this cohort of students represented a group of self-selected participants who had opted to take an online course focused on diverse learners, and therefore these participants may reflect a greater level of comfort with remote learning and teaching than the average teacher and, likewise, a greater investment in issues of language diversity, cultural diversity, and equity. Finally, this study utilized analysis of journal prompts and interviews, attempting to triangulate multiple sources to ensure greater depth and quality of data, but findings are limited to those easily conveyed in these formats. This study did not employ supplemental survey data, teaching observations, or analysis of participant performance to capture participants' knowledge, comfort with remote learning, use of effective strategies while teaching remotely, or general teaching ability. Because of these limitations, study findings are not necessarily generalizable to all populations of teacher trainees or even teacher trainees pursuing licensure in ESOL or multilingual teaching.

### 4.4. Connections to Research

It continues to appear that universally designed instruction for MLs is an area of focus for these participants, corroborating findings from prior research (both pre-COVID and post-COVID) (Flanagan and Morgan 2021; Basham et al. 2020). As participants considered their skill sets for providing instruction, they also reflected on their technological proficiency and available resources. The focus on technological resources (such as Wi-Fi or apps) may be an area for districts and systems to continue to explore with the goal of ensuring equal access for all populations, especially language learners.

### 4.5. Conclusions

This study identified patterns and themes in the perceptions of teacher trainees regarding remote learning for multilingual learners. Teacher trainees identified challenges and drawbacks in the implementation of remote learning, particularly for MLs. These

impacted student learning as well as ability to form relationships. At the same time, teacher trainees identified potential strategies and practices to improve student experiences in remote learning. These findings are relevant to the future preparation of teachers, who are increasingly expected to integrate technology and virtual learning experiences into teaching. They are relevant as well to the specific population of MLs, who may experience transition between educational settings, increased mobility, or educational interruptions and may benefit from remote or virtual learning opportunities in those contexts. Further exploration of these topics in research, particularly ways to support those training for certification or endorsement as English language teachers in building relationships with students and families, instructional strategies for engaging learners and families in virtual settings, and integration of UDL-based practices into virtual teaching, is warranted. These topics may be appropriate to consider in further research and may be useful to integrate into teacher education programs as well as professional development for future teachers of language learners. While teacher trainees of MLs typically cover technology and relationship-building in passing, if not in full courses, these topics may merit sustained and dedicated attention as they relate to the virtual teaching setting, particularly as the use of virtual and technology-assisted modalities continues to become more frequent. Likewise, concepts such as effective instruction, differentiation, and appropriate selection of strategies may be addressed as they relate to the needs of MLs in a technology-assisted or online environment as well as the more traditional face-to-face classroom.

**Author Contributions:** Conceptualization, K.M.H., G.K. and P.R.D.; methodology, K.M.H. and G.K.; data collection, H.H., G.K. and P.R.D.; data analysis, P.R.D., H.H., K.M.H. and G.K.; resources: K.M.H. and G.K.; writing—original draft preparation, K.M.H. and P.R.D.; writing—review and editing, K.M.H., G.K., H.H. and P.R.D.; funding acquisition, P.R.D. All authors have read and agreed to the published version of the manuscript.

**Funding:** This work was supported by the U.S. Department of Education, Office of English Acquisition under Grant NPD.2017.T36Z170189. The views expressed herein do not necessarily represent the positions or policies of the Department of Education. No official endorsement by the U.S. Department of Education of any product, commodity, service, or enterprise mentioned within this manuscript is intended or should be inferred.

**Institutional Review Board Statement:** The study was approved by the Institutional Review Board of Towson University (Protocol Number: #1804034177; initial approval 13 June 2019; annual approval 6 May 2021).

**Informed Consent Statement:** Informed consent was obtained from all subjects involved in the study.

**Data Availability Statement:** The data presented in this study are available on request from the corresponding author. The data are not publicly available due to student privacy considerations.

**Acknowledgments:** The authors appreciatively acknowledge administrative support from Laura Cometa and Lacey Rupp in the preparation of this article.

**Conflicts of Interest:** The authors declare no conflict of interest. The funders had no role in the design of the study; in the collection, analyses, or interpretation of data; in the writing of the manuscript, or in the decision to publish the results.

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
