# Peer review of "Teacher Trainees’ Perspectives on Remote Instruction for Multilingual Learners of English"

_languages, doi:10.3390/languages8010085_

Round 1

Reviewer 1 Report

  1. The sample size is too small, only 9 participants selected from ‘a cohort of 20 pre-service teachers from varied fields’. Additionally, the diversity of the participants (‘Early Childhood, Elementary Education, Special Education, and Elementary Education-Special Education’) and their small number makes it difficult to compare the research findings based on their self-perceived experiences as each field can require very different strategies depending on the target students (age, level, needs, etc). Besides, no explanation is provided about the inclusion of Participant 3 ‘Social Sciences and Secondary Education (ESOL endorsement)’, the only one in Secondary Education, which was not originally stated to be part of the cohort. Therefore, their experiences may have been very diverse depending on their field but this is not properly analysed in the paper. Alternatively, it might have been more interesting to focus on one specific group/field (Special Education, for example) and the participants’ experience.
  2. The approach and methodology are not well explained. The phenomenological perspective mentioned in lines 132-133 is not described and related with the analysis employed in this paper. The statement that the ‘study used a descriptive-interpretive qualitative methodology’ needs further clarification. No qualitative data analysis software has been used, so most of this paper is based on the self-perceived experiences of pre-service teachers and the (verbatim? manual?) transcriptions obtained from the journals and/or interviews. A large part of this paper basically copies of reproduces what participants allegedly wrote in their journal or said in the interview, which is not made clear. Unfortunately, no correlation is established between the transcribed statements of each participant and their corresponding field or profile (Special Education, Elementary, etc). This means that the research findings are based on the self-perceived experiences of 9 participants with very different fields as transcribed/reported by the author/s.
  3. Three of the four research questions are not well analysed. Only the first one about participants’ experience is clearly exposed. The second (challenges) and third (benefits) are not clearly specified and the last one (useful strategies or approaches) is not exemplified. Given the diversity of the participants it might be difficult to correlate the experiences of a pre-service teacher in Early childhood (participants 8 and 9) with the participant in Secondary Education (participant 3).
  4. Consequently, most of the research findings are ‘as evident as they can be’ regarding challenges and benefits. No specific details or examples of ‘useful strategies’ are provided apart from the occasional reference to traditional tools such as Powerpoint or Kahoot. Therefore, research question number 4 (useful strategies) is not analysed.
  5. In the ‘Data collection and Analysis’ section the author/s need to correlate the prompts they mention with the objectives. Each prompt contains many questions and there is no clear indication about their purpose and correlation with the research objectives previously mentioned.
  6. No details are provided about the participants’ journal or the interviews (how long?, when? Were they recorded/transcribed?, etc)
  7. The method needs to be improved. For example, ‘The research team reviewed all transcripts and journal entries. The research team engaged in multi-level coding (Saldana 2015) in which key codes were first identified, then consolidated into categories, reviewed and interpreted as general themes were identified within and across topics and questions’ (227-230). This statement poses many questions. Who was this research team? the author/s? Selection criteria of its members? How was the coding performed if no qualitative data analysis software was used? How were the general themes interpreted? Was the research team also responsible for conducting the interviews? Were all participants interviewed together or separately? The research methodology needs to be improved and clarified.
  8. In lack of a more reliable qualitative analysis, some of the alleged challenges/benefits are repeated in different sections and are too vague (for example ‘adjustment’, ‘flexibility’). What do they specifically refer to depending on the participants’ profile and student needs (time adjustment, technological adjustment, Special Education adjustment?). They seem to be too general and evident. Unsurprisingly, some statements seem to refer to these issues such as ‘Challenges with technology also extended to concerns about the ability of different age groups to utilize technology successfully (324-325)’ but a deeper analysis is necessary depending on participants’ profile and target students.
  9. The results section (participants’ transcriptions) is too long and already includes discussion. Comparatively, the last section entitled ‘Discussion and Conclusions’ is too short and no clear conclusions are provided apart from some evident statements. The author/s should consider clearly separating the results section (participants’ transcription: journal and interview) from the discussion and perform a deeper analysis based on the research findings.
  10. Finally, the paper needs to include the research limitations, which are many as previously exposed. No indication has been included.

This paper is mostly based on the self-perceived experience of a very small number of pre-service teachers (9) with very diverse profiles (Early Childhood, Elementary, Special, Secondary?). The research findings (participants’ transcriptions) need to be specified as they are too ‘obvious. The method needs to be better explained.

Author Response

Thanks very much for this extensive and detailed feedback. Responses to each item are below. We appreciate your time and willingness to review and we thank you for this candid feedback which we believe has helped to strengthen our paper significantly. 

  1. The sample size is too small, only 9 participants selected from ‘a cohort of 20 pre-service teachers from varied fields’. Additionally, the diversity of the participants (‘Early Childhood, Elementary Education, Special Education, and Elementary Education-Special Education’) and their small number makes it difficult to compare the research findings based on their self-perceived experiences as each field can require very different strategies depending on the target students (age, level, needs, etc). Besides, no explanation is provided about the inclusion of Participant 3 ‘Social Sciences and Secondary Education (ESOL endorsement)’, the only one in Secondary Education, which was not originally stated to be part of the cohort. Therefore, their experiences may have been very diverse depending on their field but this is not properly analysed in the paper. Alternatively, it might have been more interesting to focus on one specific group/field (Special Education, for example) and the participants’ experience. While we retained the focus on multiple groups (early childhood education, special education, etc.) we did add clarification and description about the participants' backgrounds as well as the methodology. 
  2. The approach and methodology are not well explained. The phenomenological perspective mentioned in lines 132-133 is not described and related with the analysis employed in this paper. The statement that the ‘study used a descriptive-interpretive qualitative methodology’ needs further clarification. No qualitative data analysis software has been used, so most of this paper is based on the self-perceived experiences of pre-service teachers and the (verbatim? manual?) transcriptions obtained from the journals and/or interviews. A large part of this paper basically copies of reproduces what participants allegedly wrote in their journal or said in the interview, which is not made clear. Unfortunately, no correlation is established between the transcribed statements of each participant and their corresponding field or profile (Special Education, Elementary, etc). This means that the research findings are based on the self-perceived experiences of 9 participants with very different fields as transcribed/reported by the author/s. We have added information to clarify the methodology, particularly the analysis and use of software in that process. We have also referenced the study's small sample size as a limitation of the study. 
  3. Three of the four research questions are not well analysed. Only the first one about participants’ experience is clearly exposed. The second (challenges) and third (benefits) are not clearly specified and the last one (useful strategies or approaches) is not exemplified. Given the diversity of the participants it might be difficult to correlate the experiences of a pre-service teacher in Early childhood (participants 8 and 9) with the participant in Secondary Education (participant 3). We provided more context about participants' backgrounds as related to the methodology section. We clarified responses and results where possible and specified reasons (such as participants' lack of experience/ lack of familiarity with specific strategies) where possible. 
  4. Consequently, most of the research findings are ‘as evident as they can be’ regarding challenges and benefits. No specific details or examples of ‘useful strategies’ are provided apart from the occasional reference to traditional tools such as Powerpoint or Kahoot. Therefore, research question number 4 (useful strategies) is not analysed. We have provided more detail regarding participants' clarification of strategies, including strategies for relationship building engagement as well as purposeful use of technology. 
  5. In the ‘Data collection and Analysis’ section the author/s need to correlate the prompts they mention with the objectives. Each prompt contains many questions and there is no clear indication about their purpose and correlation with the research objectives previously mentioned. We have clarified the role of the prompts and connected them to the research more clearly. 
  6. No details are provided about the participants’ journal or the interviews (how long?, when? Were they recorded/transcribed?, etc). We have provided more detail about the journal and interview data.
  7. The method needs to be improved. For example, ‘The research team reviewed all transcripts and journal entries. The research team engaged in multi-level coding (Saldana 2015) in which key codes were first identified, then consolidated into categories, reviewed and interpreted as general themes were identified within and across topics and questions’ (227-230). This statement poses many questions. Who was this research team? the author/s? Selection criteria of its members? How was the coding performed if no qualitative data analysis software was used? How were the general themes interpreted? Was the research team also responsible for conducting the interviews? Were all participants interviewed together or separately? The research methodology needs to be improved and clarified. We have strengthened and added to the discussion of these elements in the methodology. 
  8. In lack of a more reliable qualitative analysis, some of the alleged challenges/benefits are repeated in different sections and are too vague (for example ‘adjustment’, ‘flexibility’). What do they specifically refer to depending on the participants’ profile and student needs (time adjustment, technological adjustment, Special Education adjustment?). They seem to be too general and evident. Unsurprisingly, some statements seem to refer to these issues such as ‘Challenges with technology also extended to concerns about the ability of different age groups to utilize technology successfully (324-325)’ but a deeper analysis is necessary depending on participants’ profile and target students. We have clarified these issues and added some deeper analysis of participants' responses.
  9. The results section (participants’ transcriptions) is too long and already includes discussion. Comparatively, the last section entitled ‘Discussion and Conclusions’ is too short and no clear conclusions are provided apart from some evident statements. The author/s should consider clearly separating the results section (participants’ transcription: journal and interview) from the discussion and perform a deeper analysis based on the research findings. We have restructured and extended the results and discussion sections, adding to the discussion section and moving some content to that section, to address this concern.
  10. Finally, the paper needs to include the research limitations, which are many as previously exposed. No indication has been included. We have included a limitations section to address this concern.

Reviewer 2 Report

While the study provides some valuable insights into the world of remote teaching from the perspective of prospective teachers, there are a number of issues that need to be addressed to make it a solid paper to be published.

1) The title focuses on supporting learners, but that is not what the study or the paper are about. The study is about preservice teachers' perspectives on remote learning.

2) There is no need to use "multilingual learners" in the study. The group is clearly English language learners (ELLs). Using multilingualism here would require an exploration of multilingualism, starting with providing these learners' linguistic backgrounds. Whether or not this data is available, you need to describe who the preservice teachers were teaching or tutoring. This is connected to the following point.

3) You are describing preservice teachers' perspectives, but we as readers do not know what they were doing - who were they teaching? You sometimes talk about tutoring and sometimes about teaching. Did they work one-on-one exclusively, if yes, with who? Who were their students? Did they work with whole classes? You mention small groups? Small group of what kinds of learners? This is important information as it surely influences the preservice teachers' experiences.

4) I was doing the math but I still do not know how many participants you had. You mention a cohort of twenty but then provide data for nine and you also mention that all but one participated in the interview. The number of participants should be clearly stated in the methods section and the abstract.

5) The abstract should also contain information about the context, i.e. the country in which the study was carried out.

6) It is confusing when in the abstract and the paper you talk about students - sometimes to refer to preservice teachers and sometimes to refer to their students. Please use it only for the latter and use preservice teacher, teacher candidates (as you do in one part), and any other term that would make it less confusing than using "student" for both groups. For example, in the abstract in the methods part you talk about preservice teachers, and in the results part about students, so it is not immediately clear that the latter are actually the former.

7) A minor note - "robust, culturally and linguistically responsive learning opportunities" sounds quite odd. Can learning opportunities be "robust"?

8) I am not sure what is meant by the following sentence: "Hamilton et al. (2020) also explored teacher practices and family engagement during the pandemic to identify areas for focus and emphasis in school practices" as it does not tell us what Hamilton et al. found, just that they carried out a study.

9) I am not sure what the point of 1.1. Framework is - what is its connection to the paper and study at hand? It seems like something that could easily be left out as you do not refer to it later on.

10) You mention universal design for learning several times, but you need to provide more information about it as it is not universally familiar.

11) In the theoretical part you briefly address student concerns with remote learning, but for your paper it would be much more relevant and important to address and research regarding the challenges that teachers face in remote teaching.

12) In the part where you show the results for "Question 1: How do preservice teachers describe their experience as a learner within a virtual experience in the COVID-19 pandemic?" it is confusing that some of the answers refer to the preservice teachers' experiences as learners (e.g. 3.2.2. and 3.2.3.) and some as teachers (e.g. 3.2.1. and 3.2.5.) - why is this the case if the question was formulated clearly? The question aims to investigate their perspectives as learners, not teachers. Can you explain what went wrong?

13) In the presentation of the results, it is not clear what data comes from the journal and what from the interview. This should be stated or explained.

14) In your discussion, you should explicitly come back to the research questions and answer them.

15) It would be interesting to compare the results with other studies on teachers' experiences. Do preservice teachers face similar challenges as more experienced ESL teachers? Do they have unique challenges? This is connected to note 11 - you need to refer to current studies on teachers' experiences with remote teaching, preferably ESL or EFL.

16) Finally, since this is a study about preservice teachers' perspectives, or rather, experiences, I feel it is important to know more about them - did they teach from home or the institution, were they good with technology, did they have their own computer, etc. I understand this data may be impossible to collect at this point, but I am just highlighting it as important when describing someone's remote teaching experience.

The paper provides valuable insights and certainly has potential, but the notes above need to be addressed in order for the paper to be published.

Author Response

Thank you for your candid and helpful feedback. We appreciate it and the time you took to review our paper. We have responded, below, to the comments in your review, which we believe have helped to strengthen our paper. Thank you again. 

1) The title focuses on supporting learners, but that is not what the study or the paper are about. The study is about preservice teachers' perspectives on remote learning. We have updated the title to better reflect this focus. 

2) There is no need to use "multilingual learners" in the study. The group is clearly English language learners (ELLs). Using multilingualism here would require an exploration of multilingualism, starting with providing these learners' linguistic backgrounds. Whether or not this data is available, you need to describe who the preservice teachers were teaching or tutoring. This is connected to the following point. Respectfully, we have retained for now the use of "multilingual learners" as this term is increasingly preferred as a way to indicate the variety of language skills which many language learners bring to their schooling. We are open to continued revisions based on the preference of the editorial team. 

3) You are describing preservice teachers' perspectives, but we as readers do not know what they were doing - who were they teaching? You sometimes talk about tutoring and sometimes about teaching. Did they work one-on-one exclusively, if yes, with who? Who were their students? Did they work with whole classes? You mention small groups? Small group of what kinds of learners? This is important information as it surely influences the preservice teachers' experiences. While preservice participants had experienced a range of teaching experiences, we have attempted to clarify this. 

4) I was doing the math but I still do not know how many participants you had. You mention a cohort of twenty but then provide data for nine and you also mention that all but one participated in the interview. The number of participants should be clearly stated in the methods section and the abstract. This has been updated and clarified. 

5) The abstract should also contain information about the context, i.e. the country in which the study was carried out. The abstract has been updated to provide this. 

6) It is confusing when in the abstract and the paper you talk about students - sometimes to refer to preservice teachers and sometimes to refer to their students. Please use it only for the latter and use preservice teacher, teacher candidates (as you do in one part), and any other term that would make it less confusing than using "student" for both groups. For example, in the abstract in the methods part you talk about preservice teachers, and in the results part about students, so it is not immediately clear that the latter are actually the former. We have attempted to use the term "participants" or "preservice teachers/ candidates" for that group and eliminate the general use of "students" to describe participants in the study.

7) A minor note - "robust, culturally and linguistically responsive learning opportunities" sounds quite odd. Can learning opportunities be "robust"? We have reviewed and addressed this word choice. 

8) I am not sure what is meant by the following sentence: "Hamilton et al. (2020) also explored teacher practices and family engagement during the pandemic to identify areas for focus and emphasis in school practices" as it does not tell us what Hamilton et al. found, just that they carried out a study. We have clarified this sentence. 

9) I am not sure what the point of 1.1. Framework is - what is its connection to the paper and study at hand? It seems like something that could easily be left out as you do not refer to it later on. We have restructured and retitled this section. 

10) You mention universal design for learning several times, but you need to provide more information about it as it is not universally familiar. We have extended our discussion of UDL in the body of the literature review. 

11) In the theoretical part you briefly address student concerns with remote learning, but for your paper it would be much more relevant and important to address and research regarding the challenges that teachers face in remote teaching. We have added research to address this point to our literature review. 

12) In the part where you show the results for "Question 1: How do preservice teachers describe their experience as a learner within a virtual experience in the COVID-19 pandemic?" it is confusing that some of the answers refer to the preservice teachers' experiences as learners (e.g. 3.2.2. and 3.2.3.) and some as teachers (e.g. 3.2.1. and 3.2.5.) - why is this the case if the question was formulated clearly? The question aims to investigate their perspectives as learners, not teachers. Can you explain what went wrong? We have attempted to clarify this issue in our paper. We also believe that some participants, having recent experiences as virtual learners, brought this perspective to bear in their responses insofar as that experience could help them to better understand the process of teaching. We have attempted to clarify this in the body of the paper. 

13) In the presentation of the results, it is not clear what data comes from the journal and what from the interview. This should be stated or explained. We have added to the methodology section to clarify this. 

14) In your discussion, you should explicitly come back to the research questions and answer them. We have clarified the structure of the paper to make this clear. 

15) It would be interesting to compare the results with other studies on teachers' experiences. Do preservice teachers face similar challenges as more experienced ESL teachers? Do they have unique challenges? This is connected to note 11 - you need to refer to current studies on teachers' experiences with remote teaching, preferably ESL or EFL. We have added some reference to this literature and have strengthened the literature review generally to incorporate more literature about preservice teachers of ESOL/ ESL.

16) Finally, since this is a study about preservice teachers' perspectives, or rather, experiences, I feel it is important to know more about them - did they teach from home or the institution, were they good with technology, did they have their own computer, etc. I understand this data may be impossible to collect at this point, but I am just highlighting it as important when describing someone's remote teaching experience. Thank you for this feedback. Not all of this data was available as it was not all collected, but we have clarified information about the participants where possible and have noted this for design of future studies in this area. 

Reviewer 3 Report

I think the “Background” needs a bit of work since the “subsections” : “methods” “results”, “conclusions” break the coherence of the text.

The literature review is very “choppy” consisting essentially of a list of references regarding issues in online learning. Synthesis of the literature is quite poor. Moreover, most of the literature review focuses on the issues of online learning during the pandemic for school students and not on university/higher education teacher education programmes.I think that a discussion of Universal Design is necessary.

The methodology of the study has been well described. Perhaps the sampling techniques used could be condensed and some more information on the interview questions is needed. In essence the study reports on the responses of 9 participants. It is not clear in the presentation of results how many students answered what.

Also the presentation of results is quite “choppy” as well. There are too many sections and subsections which obstruct the flow of the paper. More quotations from the students are necessary.

The discussion is well written and relates to the research questions set.

Author Response

Thank you for your time and dedication in providing this feedback, which we believe has helped us to strengthen this paper. Below, we respond to the points noted in your feedback. We appreciate, again, your effort and detailed review.

The literature review is very “choppy” consisting essentially of a list of references regarding issues in online learning. Synthesis of the literature is quite poor. Moreover, most of the literature review focuses on the issues of online learning during the pandemic for school students and not on university/higher education teacher education programmes.I think that a discussion of Universal Design is necessary. We have added a discussion of Universal Design and teacher preparation and extended the literature review generally to address this concern. 

The methodology of the study has been well described. Perhaps the sampling techniques used could be condensed and some more information on the interview questions is needed. In essence the study reports on the responses of 9 participants. It is not clear in the presentation of results how many students answered what. We have added detail on the interview and sampling techniques. 

Also the presentation of results is quite “choppy” as well. There are too many sections and subsections which obstruct the flow of the paper. More quotations from the students are necessary. We have included quotations from students where feasible and have condensed our presentation of results, moving some subsections to the discussion section, to address this concern. 

The discussion is well written and relates to the research questions set.

Round 2

Reviewer 1 Report

Dear authors,

Thanks for considering the recommendations. I believe your paper now is almost ready for publication.

Just two  details related with your paper structure:

1. Section :  '3. Results and Discussion Findings(?)' 

*question mark is yours in the revised version

In rigorous scientific works, 'research data'  refer to quantitatite results  whereas 'research findings' refer to qualitative results. Based on your analysis method (NVIVO)  yours would probably fit more into 'Research findings' as the heading of this section but this is only a recommendation. Just think about the difference between 'data results' and 'research findings' throughout your paper.

2. NUMBER of sections (revise)

Your manuscript has (too) many sections, which may be confusing or even discouraging for the potential reader. The article looks more like a book with so many sections and subsections. I would suggest to keep them to the minimum. Articles are not books. For example, rename and separate section  '4. Discussion and Conclusions'. There are several (sub?)sections under this heading, the first one is numbered as '4.1 General Themes' but this is followed but other three sections which are not numbered 'Pedagogical Implications', 'Limitations' and 'Conclusions'. Why? So the question is what is '4.1' doing there if there are not 4.2. and 4.3 sections? And why do you name it '4. Discussion and Conclusions' first and then incude another section entitled 'Conclusions' with no number?

I am afraid you tried to include too many sections and subsections and you got confused. I believe this can be also very confusing and discouraging for the potential reader. Therefore, I would SIMPLIFY the structure,  separate your last sections into 3:

-'Results'  or  'Research Findings' (that is what they are -qualitative)

-'Discussion' (or 'Discussion and Implications' if you prefer)

-and  'Conclusions' (you may include limitations and further research here, no need to separate them).

This would be an easy way to (re)organize the paper but this is your decission (style) and beware that different reviewers may hold different opinions. Just think about the potential readers and the journal instructions in this sense. Make sure people can easily read the paper without getting lost in all these numbers and sections.

Author Response

Thank you for this careful and thoughtful feedback! We have adjusted and streamlined the section headings and attended to the wording comment below as well. We appreciate your time and review of this revision!

Reviewer 2 Report

The manuscript has been significantly improved. Well done!

Author Response

Thank you again for your careful and thoughtful feedback.